# Polyphosphazene-Mediated Assembly of TLR4 and TLR7/8 Agonists Enables a Potent Nano-Adjuvant Delivery System for Hepatitis C Virus Vaccine Antigens

**DOI:** 10.3390/vaccines13101012

**Published:** 2025-09-28

**Authors:** Alexander K. Andrianov, Alexander Marin, Sarah Jeong, Liudmila Kulakova, Ananda Chowdhury, Raman Hlushko, Sayan Das, Francesca Moy, Eric A. Toth, Robert K. Ernst, Thomas R. Fuerst

**Affiliations:** 1Institute for Bioscience and Biotechnology Research, University of Maryland, Rockville, MD 20850, USA; amarin1@umd.edu (A.M.); sjeong21@umd.edu (S.J.); kulakova@umd.edu (L.K.); ananda.d22@gmail.com (A.C.); hlushkor@umd.edu (R.H.); eatoth@umd.edu (E.A.T.); 2Department of Microbial Pathogenesis, School of Dentistry, University of Maryland, Baltimore, MD 21021, USAfgardner@umaryland.edu (F.M.); rkernst@umaryland.edu (R.K.E.); 3Department of Cell Biology and Molecular Genetics, University of Maryland, College Park, MD 20742, USA

**Keywords:** Hepatitis C virus, vaccine adjuvants, polyphosphazenes, vaccine delivery, Toll-like receptor agonist, intermolecular interactions

## Abstract

**Background**: The quest for well-defined immunoadjuvants remains one of the highest priorities for the successful development of effective vaccines. Combination adjuvants, which are designed to integrate both the ability to activate a variety of immune mechanisms and synergistically improve the delivery of vaccine components, are well-positioned to address the unmet needs. The development of a preventive vaccine against hepatitis C virus (HCV)—a major public health concern—is a particular instance in which the choice of the immunoadjuvant is of utmost importance. **Methods**: We assembled a lipid A Toll-like receptor 4 (TLR4) agonist BECC438 and TLR7/8 agonist resiquimod (R848) on a polyphosphazene macromolecule (PCPP) to create a nanoscale immunoadjuvant-vaccine delivery system: PCPP-R+BECC438. This aqueous-based system was formulated with the HCV sE2 antigen, and the resulting vaccine candidate was evaluated in vivo for the ability to induce immune responses. **Results**: Co-assembly of adjuvants resulted in a visually clear aqueous system of nanoscale dimensions, monomodal size distribution, and entropy-driven interactions between components. Intramuscular immunization of mice with HCV sE2 antigen formulated in a polyphosphazene-based nano-system induced ten-fold higher IgG and IgG2a titers than the antigen adjuvanted with BECC438 alone. PCPP-R+BECC438 formulated HCV sE2 also produced statistically significant improvements in IgG2a/IgG1 ratio and more robust HCVpp neutralization ID_50_ titers than control formulations. **Conclusions**: Polyphosphazene-assembled adjuvant nano-system promotes in vivo immune responses of enhanced quantity and quality of antibodies with increased potency of HCV neutralization.

## 1. Introduction

Immunoadjuvants are important components of vaccine formulations, which act by enhancing and modulating immune responses to vaccine antigens [1,2,3,4,5]. Immunization using purified protein antigens alone often elicits weak antibody responses and minimal T cell activation, requiring multiple booster doses to provide protection. Therefore, the generation of a toolbox of adjuvants with well-defined profiles, which can be applied to new vaccines against diverse pathogens, is one of the highest priorities in the field. Accordingly, adjuvants are typically classified as immunostimulating molecules or vaccine delivery systems, or a combination of the two, based on their mechanism of action. Immunostimulating molecules mainly activate the innate immune receptors that sense highly conserved pathogen-associated molecular patterns (PAMPs) [4]. In particular, once activated, Toll-like receptors (TLRs), commonly found on dendritic cells, generate signals that drive the proper activation of downstream adaptive immune responses. Vaccine delivery systems are designed to improve the stability and presentation of the vaccine to the immune system. Combination adjuvants integrate both components that act synergistically by activating a variety of immune mechanisms and improving their delivery. TLR4 agonists, such as lipid A-based molecules, are among the most potent immunostimulating compounds [6]. Of note, 3-O-desacyl-4′-monophosphoryl lipid A (MPL) molecule is a component of AS01 and AS04 adjuvant systems currently employed in licensed vaccines [4].

Monophosphoryl lipid A (MPL), known for enhancing Th1 humoral and cell-mediated immune responses to protein antigens, is currently being used in numerous vaccine trials and was licensed by the Food and Drug Administration (FDA) as part of Shingrix and Cervarix vaccines [7,8]. MPL is produced from *Salmonella minnesota* R595 lipid A using a multi-step detoxification process. Instead of the chemical treatments employed in the generation of MPL, a different approach was undertaken in the synthesis of MPL-like molecules termed Bacterial Enzymatic Combinatorial Chemistry (BECC). Using heterologous expression of various acyltransferases, phosphatases, deacylases, and glycosyltransferases in different bacterial backgrounds, we have been able to generate numerous lipid A structures, such as BECC438, with altered proinflammatory responses in humans and murine cells [9]. BECC lipid A molecules have proven potent adjuvant capabilities and, due to their lipophilic nature, present opportunities for innovative formulation strategies to improve their stability in aqueous dispersions [10]. These potent adjuvants can benefit from delivery vehicles, such as water-soluble polymers, emulsions, liposomes, and nanoparticles, for optimal formulation stability, effective display of TLR4 ligands, and association with vaccine antigens.

Polyphosphazene immunoadjuvants are synthetic biodegradable macromolecules, which are composed of an inorganic phosphorus–nitrogen backbone and organic side groups containing anionic functionalities [11]. Polyphosphazenes are known to display intrinsic immunostimulatory activity both in vitro and in vivo [12,13,14]. However, perhaps an even more important feature of these macromolecules is their role as vaccine delivery vehicles. This enabling functionality is largely supported by their remarkable ability to spontaneously self-assemble with vaccine antigens in aqueous solutions [15,16]. The resulting supramolecular complexes are of nano-scale dimensions and may be viewed as virus-mimicking structures in terms of their size and ability to present protein antigens [17]. Poly[di(carboxylatophenoxy)phosphazene] (PCPP) is an advanced polyphosphazene adjuvant, which has demonstrated safety and potency in several clinical trials [18,19,20,21,22]. In addition to antigen carrying capacity, PCPP can effectively bind Toll-like receptor (TLR) agonists, such as resiquimod, via counterion–polyelectrolyte interactions to both extend their half-life and provide association with the antigen [23]. The capacity of this macromolecule to bind to and transport other immunostimulating molecules remains understudied.

Hepatitis C virus (HCV), a global public health concern which causes infections in over fifty million people worldwide, is also a major cause of liver disease and cancer [24,25,26]. HCV infection is known as the “silent epidemic” because many people with the disease do not know they are infected. In the United States, more than 3.2 million Americans are infected, and yet, as many as 75% are unaware that they carry the virus [27]. Over the past decade, highly effective direct-acting antivirals (DAAs) have been developed that can nearly eliminate the virus in HCV-infected individuals. However, these DAAs cannot provide protection against reinfection and associated disease later in life, and their availability to people in developing countries is severely limited [28,29,30]. Hence, the World Health Organization (WHO) has developed a global strategy with a goal of bringing down new HCV infections by ninety percent and related deaths by sixty-five percent by the year 2030 [31]. Development of an effective vaccine is a critical step in addressing this major public health problem.

Overcoming the genetic diversity of HCV is a major hurdle in the development of an effective vaccine, with eight known genotypes and over one hundred subtypes identified to date [32]. The diversity of the virus is due to the lack of a proof-reading function of the virally encoded RNA-dependent RNA polymerase, leading to the extremely high sequence variability. As a result, the amino acid median sequence divergence within and between genotypes is approximately 23% and 33%, respectively [33]. Given this diversity, a prophylactic vaccine must be capable of inducing broadly reactive immune responses that can provide protection against a global pool of circulating viruses. Moreover, results from both macaque and human studies have shown that the induction of HCV-specific CD4+ and CD8+ T cells are critical in curtailing the infection [34,35]. Hence, an effective vaccine should be capable of eliciting both humoral and cellular responses in which the choice of the immunoadjuvant and delivery system is of utmost importance.

In this study, we combined BECC438, a lipid A TLR4 agonist, with PCPP or PCPP-resiquimod (PCPP-R) to derive binary adjuvants PCPP+BECC438 and PCPP-R+BECC438, respectively, that remain in an aqueous state. Here, PCPP scaffold served as the solubilizing excipient for BECC438, and stability was imparted via lyophilization, such that upon resuspension in PBS, the binary adjuvant remained visually clear and retained a nano-scale monomodal size distribution similar to PCPP. We tested the immunostimulatory effects of the binary adjuvant in vitro and in mice with HCV-sE2 antigen. Using immortalized mouse macrophages, the BECC438 component of the binary adjuvant retained immunostimulatory activity implying no antagonistic effect from complexing with PCPP, and at higher concentrations the binary adjuvant showed somewhat superior activity implying a possibility for dose-dependent synergy with further ratio optimizations. Intramuscular immunization of mice with HCV sE2 antigen formulated in PCPP-R+BECC438 (or PCPP-R) produced 10-fold higher IgG titers and IgG2a titers than BECC438 alone. PCPP-R+BECC438 also produced statistically significant differences in IgG2a/IgG1 ratio and more robust HCVpp neutralization ID50 titers that were two to three times higher than BECC438 and PCPP-R, respectively. Our experiments demonstrate that a combination of the novel BECC438 adjuvant with a water-soluble PCPP-based delivery system led to the induction of enhanced antibody production with increased potency of HCV neutralization.

## 2. Materials and Methods

### 2.1. Protein Expression and Purification

To construct the expression plasmid for sE2, the sequence encoding gt1a isolate H77C (GenBank accession number AF011751) residues 384 through 661 with the addition of a C-terminal hexahistidine tag was subcloned into the mammalian expression vector pSecTag2 (Invitrogen). The protein sE2 was transiently expressed using the Expi293 Expression System and the protocol provided by the manufacturer (ThermoFisher, Waltham, MA, USA). The cells were cultured in a shaking incubator at 37 °C, 120 rpm, and 8% CO_2_ in Expi293 medium until reaching a cell density of 2.0 × 10^6^ cells/mL. The transfection mixture contained a predetermined amount of plasmid, ExpiFectamine™ 293 Reagent and other transfection enhancers. The sE2 supernatant was harvested 72 h after transfection then centrifuged in order to remove the suspended cells at 10,000 rpm for 10 min and filtered using 0.22 μm filters. The protein was then purified from the supernatant using sequential HisTrap Ni2+-NTA and Superdex 200 size-exclusion chromatography as described previously [36,37].

### 2.2. Preparation of BECC438

BECC438b was made and resuspended as previously described [9]. Briefly, BECC438 was grown in shaking culture at 26 °C for 18 h, during which time the culture reached an OD600 of 1.0–1.4. After pelleting bacteria from the liquid culture, lipooligosaccharide (LOS) was extracted from the pellet as previously described using the double hot phenol method [38]. Mass spectrometry was used to confirm the extracted lipid A structures (Bruker Microflex MALDI TOF, Billerica, MA, USA, norharmane matrix, negative ion mode). The resulting LOS was designated BECC438b (b indicating biologically derived).

### 2.3. Preparation of PCPP-BECC438 Formulations

Aqueous-based formulations of PCPP and BECC438 were prepared at PCPP and BECC438 concentrations of 0.5 and 0.25 mg/mL, respectively. Lyophilized powder of BECC438 was suspended in endotoxin-free deionized water containing 0.02% triethanolamine (TEA) (Millipore Sigma, St. Louis, MO, USA) to make a 0.5 mg/mL stock solution. The dispersion was vortexed and then sonicated in a Branson CPX2800H ultrasonic cleaner (Emerson Electric Co., St. Louis, MO, USA) for approximately 10 min at room temperature. The obtained micellar solution of BECC438 was mixed at a 1:1 (*v*/*v*) ratio with 1 mg/mL aqueous solution of PCPP (pH 7.4). Formulations were then lyophilized and re-suspended to a desirable concentration in a phosphate-buffered saline (PBS) (Thermo-Fisher Scientific, Grand Island, NY, USA) at pH 7.4.

### 2.4. Dynamic Light Scattering (DLS) and Isothermal Titration Calorimetry (ITC) Analysis

DLS characterization of BECC438, PCPP, and their lyophilized formulation re-dissolved in PBS (pH 7.4) was conducted using a Malvern Zetasizer Nano ZS instrument with data analysis carried out with version 7.10 Zetasizer Software (Malvern Instruments Ltd., Worcestershire, UK).

ITC measurements were conducted using a Nano ITC SV instrument (TA Instruments, Waters, New Castle, DE, USA) in an aqueous solution (50 mM phosphate buffer, pH 7.5) at 25 °C. BECC438 (2.0 mg/mL) was dispersed in an aqueous phase and placed into an isothermal cell of the instrument. Titration was carried out using a 250 μL syringe rotating at 200 rpm (5.0 mg/mL PCPP, 300 s delay between 8 μL injections). A heat release curve (μJ/s vs. s) was integrated using NanoAnalyze software, version 3.12.5 (TA Instruments, Waters, New Castle, DE, USA).

### 2.5. In Vitro Evaluation of Immunostimulating Activity

In vitro evaluation was conducted using engineered mouse macrophages—RAW BLUE cells (InvivoGen, San Diego, CA, USA) assays. Cells were maintained in culture in Dulbecco’s modified eagle medium containing glucose and L-glutamine (Thermo-Fisher Scientific, Grand Island, NY, USA) supplemented with 10% fetal bovine serum (10%), penicillin streptomycin (1%) (Thermo-Fisher Scientific, Grand Island, NY, USA), and normocin (100 µg/mL) (InvivoGen, San Diego, CA, USA). The experiment was initiated by adding ternary formulations containing PCPP, BECC438, and E2 antigens in cultures to an equal volume of RAW BLUE cells in 96 well plates (120,000 cells/well) and incubating them at 37 °C in carbon dioxide (5%) for 20 h. The level of the RAW BLUE cell activation was assessed spectrophotometrically by the conversion of SEAP substrate—p-nitrophenylphosphate (Millipore Sigma, St. Louis, MO, USA). Culture supernatant was added to the reagent at one-to-ten ratio by volume. The absorbance was read at 405 nm using a ThermoScientific SpectraMax plate reader (Molecular Devices, San Jose, CA, USA). All experiments were performed in triplicate.

### 2.6. In Vivo Studies

Six- to eight-week-old, Balb/c mice (Jackson Laboratories, ME, USA) were immunized intramuscularly (IM) in the hind, caudal thigh on day 0 (prime), followed by three boosts (days 14, 28, and 42). Fifty µL of vaccines (sE2-PCPP-R, sE2-BECC438, sE2-PCPP-R+BECC438, sE2-no adjuvant) were used for each vaccination dose. Formulations were prepared by sequential additions of R848 and sE2 to PCPP-BECC438 in an aqueous solution and were analyzed by DLS prior to injection. The following z-average hydrodynamic diameters were obtained: 63 nm (sE2-PCPP-R), 193 nm (sE2-BECC438), and 62 nm (sE2-PCPP-R+BECC438). No aggregation was observed. The control group was immunized with sterile PBS. Terminal bleeds were performed on day 56 to obtain serum. Animals were housed in individually vented cages and had access to food and water ad libitum. All animal procedures were reviewed and approved by the University of Maryland, Baltimore Institutional Animal Care and Use Committee.

### 2.7. Enzyme-Linked Immunosorbent Assay (ELISA)

ELISAs were performed to determine the HCV E2 antibody binding profile of the immunized mice sera. The plates were initially coated with 5 µg/mL Galanthus Nivalis Lectin (GNL, Vector Laboratories) and then incubated at 4 °C overnight. The subsequent day, the plates were first washed with PBST (1× PBS with Tween 20 at 0.05% *v*:*v*) and blocked with a blocking buffer (1× PBS/2% dry milk/5% FBS), and then the sE2 antigen was immobilized at a concentration of 2 μg/mL, with incubation overnight at 4 °C. The plates were then blocked in the same manner as above. Sera was initially diluted 1:100, followed by seven consecutive 5-fold serial dilutions, and then incubated on the plate for 1 h at room temperature. The antibodies were detected by a 1:5000 dilution of HRP-conjugated anti-mouse IgG secondary antibody (Abcam) with TMB substrates (Bio-Rad Laboratories). The data were collected on a SpectraMax M3 microplate reader by measuring the absorbance at 450 nm. The endpoint titers were defined as four times the highest signal elicited by the preimmune sera and determined using a curve fitting program in GraphPad Prism software version 10 (Dotatics, San Diego, CA, USA).

### 2.8. HCV Pseudoparticles (HCVpp) Neutralization Assay

The neutralizing ability of the antibodies induced by immunization of the mice was assessed with HCV pseudoparticles (HCVpp). Huh7 cells derived from human hepatoma tissue were grown in DMEM 10% FBS and 1% Pen/Strep and were seeded onto a white 96-well plate (Corning) at 1.5 × 10^4^ per well and were incubated in a CO_2_ incubator at 37 °C overnight. The next day, heat-inactivated sera or HmAbs were serially diluted and then incubated in HCVpp for 1 h. After one hour, the mixture is transferred to the plates and then incubated for 5–6 h. At the end of the incubation period, the HCVpp and antibody mixture is replaced with media. Seventy-two hours later, the plates are developed with 100 μL Bright-Glo (Promega) per well for 2 min at room temperature. The FLUOstar Omega plate reader (BMG Labtech, Ortenberg, Germany), was used to detect luminescence and the data are retrieved from the Mars software. The percentage of neutralization is calculated relative to a Relative Luminescence Units (RLUs) control, which is the HCVpp containing PBS in place of the serum. The RLU sample is a mixture containing HCVpp as well as the serum. The calculation is as follows 100*(1-(RLU sample/RLU control). The ID50, which is the dilution of sera that achieved 50% neutralization was determined with a nonlinear regression curve fitting with GraphPad Prism software version 10 (Dotatics, San Diego, CA, USA). Neutralization assays involving pseudoparticles were performed under BSL-2 conditions.

### 2.9. Statistical Analysis

The *p* values between groups IgG2a/IgG1 were determined using the nonparametric Kruskal–Wallis analysis of variance with Dunn’s multiple comparison test. *p* < 0.05 was considered significant. Statistical analyses were performed using GraphPad Prism software version 10 (Dotatics, San Diego, CA, USA).

## 3. Results

### 3.1. PCPP Enables Water-Soluble Formulations of BECC438

The approach to the development of PCPP-enabled nano-scale formulations of BECC438 involved a two-step process. First, a dispersion of lipid A molecules was prepared by mixing BECC438 with deionized water, adding triethanolamine, agitating, and sonicating the mixture (Figure 1a, step (I)). The sub-micron size dispersion of BECC438 (Figure 1b) was then admixed to an aqueous solution of PCPP as a solubilizing excipient, and the resulting formulation was stabilized and dehydrated by lyophilization (Figure 1b, step (II)). Analysis of the resuspended lyophilized material in PBS (pH 7.4) by DLS revealed a monomodal size distribution of nano-scale dimensions, which resembled the profile of unmodified PCPP (Figure 1b). The formulation was visually clear and did not contain any submicron or micron-sized aggregates as detected by DLS.

PCPP and BECC438 were studied for potential interactions in an aqueous phase using an isothermal titration calorimetry (ITC) method. Figure 2a shows raw ITC signals of PCPP addition to BECC438 (red curve and points) as well as the heat of PCPP dilution (blue curve and points). As seen from the Figure, polymer dilution generates signals that are below the baseline, indicating that this is an exothermic process. In contrast, the addition of PCPP to BECC438 results in an upward trend at the beginning of titration, which reveals a weak endothermic reaction. For further data analysis, the heats of dilution were subtracted from the heats of adsorption (Figure 2b). The resulting curve shows a positive heat of absorbance, which suggests entropy-driven interactions between PCPP and BECC438.

In vitro assessment of formulations was conducted using engineered mouse macrophages (RAW BLUE cells). These cells are derived from mouse RAW 264.7 macrophages and harbor a secreted embryonic alkaline phosphatase (SEAP) reporter construct. SEAP is inducible by NF-kB and AP-1 transcription factors that are downstream of several pattern recognition receptors (PRRs) of the innate immune system. The analysis of immunostimulatory activity of the PCPP-BECC438 combination in RAW BLUE cells showed that although PCPP was capable of augmenting the activity of the formulation partner, the activity of the binary formulation was mainly determined by the BECC438 component (Figure 3). No adverse effect of PCPP on the immunostimulatory effect of lipid A molecule was detected. Notably, at high dilutions, PCPP-BECC438 appears to show superior activity compared to its individual components.

### 3.2. Evaluation of Serological Responses and Neutralization

Based on the results of immunoactivation profiles and the DLS study, a combination PCPP-BECC438 adjuvant system was selected for in vivo studies with one adjustment to the study design. PCPP was used in its salt form with the TLR7/8 agonist resiquimod (R848). Previous studies demonstrated that although R848 was not effective in improving immune responses to HCV E2 secreted antigen (sE2) in mice due to its poor PK/PD profile, PCPP can convert it into a multimeric form, and such modified (PCPP-R) form resulted in superior performance compared to PCPP alone [23]. The addition of R848 did not affect the homogeneity of formulation or its DLS profiles. The design of in vivo studies is shown in Table 1.

To assess the immune response, mice were immunized with a formulation of sE2 antigen (refer to Table 1) in which the prime consisted of 50 μg sE2A and three boosts of 10 μg sE2 two weeks apart. A terminal bleed was taken at day 56 after the initial immunization. As shown in Figure 4a, the total IgG titers for both sE2-PCPP-R and sE2-PCPP-R+BECC438 groups are about 10-fold higher than those of BECC438 alone or the no adjuvant-sE2 group. Similarly, the isotype-specific analysis of IgG1 (T helper 2, Th2) and IgG2a (T helper 1, Th1, a surrogate marker for the induction of cellular immune responses) shows that in comparison to other groups, the sE2-PCPP-R and sE2-PCPP-R+BECC438 have superior IgG1 titers (Figure 4b) and a significantly higher IgG2a titer (Figure 4c). Since BECC438 and the no-adjuvant group showed negligible Ig2a antibodies, the elicitation of IgG2a isotype antibodies can be largely attributed to the PCPP-R component (Figure 4c). A further analysis shows that the combination adjuvant PCPP-R+BECC438 achieves a statistically significant difference in the ratio of IgG2a/IgG1 over sE2-PCPP-R (1.8-fold) and sE2-BECC438 (30-fold) using a one-way analysis of variance (ANOVA) test of sE2-PCPP-R+BECC438 versus sE2-BECC438 (*p* < 0.01) (Figure 4d). Thus, the inclusion of BECC438 in PCPP-R+BECC438 helped establish a more balanced Th1/Th2 response.

We next analyzed the serum from individual mice in each group for its ability to neutralize HCV pseudoparticles (HCVpp) made from the homologous GT1a (H77) strain from which the sE2 antigen was derived (Figure 5). Our results show that in comparison to the non-adjuvant group (sE2 only), individual mice in all adjuvant groups (PCPP-R, BECC438, and PCPP-R+BECC438) produced neutralizing responses, as measured by ID50 values (inhibitory dilution at which 50 percent neutralization is attained) with some mice in the binary adjuvant group showing higher titers over individual mice in the other groups (Figure 5a). As shown in Figure 5b, the mean HCVpp ID50 values show that the titers for the PCPP-R+BECC438 group is indeed 2–3 times higher than the BECC438 or PCPP-R group, respectively. Although not statistically significant, the neutralization responses indicate a potential synergy for the binary adjuvant with HCV sE2 antigen. We also tested the mouse sera for their ability to neutralize heterologous HCVpp, but the neutralization was weak across all groups and did not exhibit any differences across groups. This is an issue we plan to examine in a future study.

## 4. Discussion

Binary formulations of lipids and polymers are widely exploited in pharmaceutical sciences [39]. The disordered amorphous phase created by the polymer can improve the solubility of sophisticated dosage forms of poorly soluble active pharmaceutical ingredients (APIs), such as lipids. To that end, the formation of non-covalent bonds between the polymer and API is important, as such interactions inhibit the potential for nucleation and crystallization of a lipophilic molecule [39]. Furthermore, polymers impart barriers, both through steric and electrostatic stabilization of suspension, thereby preventing undesirable aggregation [40,41]. Nanosizing—an important pharmaceutical technique for reducing the size of colloidal suspensions of APIs—commonly relies on the stabilizing effect of polymers. The resulting polymer-based nanosuspensions are then converted to a solid dosage form using lyophilization or other dehydrating techniques [39,40,41]. Aqueous formulations of hydrophobic lipid A molecules—BECC438 (Figure 1a), can typically be described as sub-micron to micron-size micellar dispersions with excipient-dependent size distribution and stability. In contrast, PCPP is a large synthetic macromolecule, which forms true solutions in water with the hydrated polymer coil reaching hydrodynamic diameters in the range of 60–80 nm. Due to the presence of carboxylic acid moieties and aromatic rings in the PCPP structure (Figure 1a), this macromolecule is capable of forming supramolecular assemblies and complexes with various substrates via hydrogen bonds and hydrophobic interactions [15]. ITC results confirm the presence of interactions in the PCPP-BEC438 system. The biphasic pattern of titration, which involves exothermic and endothermic processes (Figure 2b), suggests two distinct stages in the mechanism of interactions between the polymer and the BECC438 dispersion. In the beginning, the process is exothermic, i.e., enthalpy-driven, which suggests bonding of PCPP to the polar surface of BECC438 micelles. Since the heat of interactions is relatively weak, it can be assumed that this process is dominated by the formation of hydrogen bonds and/or hydrophobic interactions. Further saturation of the system with PCPP manifests in endothermic changes. This may be attributed to a breakdown of large BECC438 micelles with the re-assembly of small adjuvant molecules along polymer chains. As this endothermic process must be driven by entropy, the results suggest that BECC438 molecules associated with flexible PCPP chains are less constrained in their mobility compared to the same molecules physically entrapped in large micelles. In this context, PCPP acts not only as a vaccine delivery vehicle due to its binding of HCV vaccine antigens demonstrated previously [15,17], but also as an efficient vehicle for displaying BECC438 lipid molecules. In vitro cellular experiments utilizing engineered immune cells demonstrated synergy between the components (Figure 3). This trend was especially pronounced for high dilutions, which are generally expected under in vivo conditions. Therefore, PCPP-enabled BECC438 formulations, which are obtained in a straightforward production process (Figure 1a), can be described as an effective nano-scale lipid-polymer system with a monomodal size distribution profile. Further nano-assembly of this adjuvant-delivery system was performed in an aqueous solution by sequential additions of the R848 adjuvant and E2 antigen. In such formulations, PCPP plays a key role in binding and co-presenting an antigenic component with TLR4 and TLR7/8 agonists—“danger signals”—which can be viewed as a virus-mimicking assembly.

Development of effective vaccines has shown that there is an increasingly important dependence on the choice of adjuvants to induce the required immune responses. Immunoadjuvants can be generally classified into two main types: vaccine delivery vehicles or immunostimulating molecules. The ongoing efforts on the development of an HCV vaccine have increasingly focused on empirical testing of immunoadjuvants in which benchmarking against other systems remains somewhat infrequent [42]. Inorganic Alhydrogel or oil-based emulsion—MF59 [43]—are among the most prevalent selections in such studies. The polyphosphazene macromolecules investigated here bring together vaccine delivery capabilities, which are accomplished by a spontaneous antigen–polymer assembly with immune stimulation manifested in the induction of a local proinflammatory response with a recruitment of immune cells to the injection site [11]. Contemporary preventive vaccines are required to engender protective immunity and persistent immunological memory, which can be achieved through the induction of both humoral and cellular responses [1]. This has led to the customized immunoadjuvant approach that can provide specific modulations of the immune response tailored to the specific chemical pattern of a pathogen, such as double-stranded RNA of viral origin (TLR3), bacterial lipopolysaccharide (TLR4), singled-stranded RNA (TLR7/8), and unmethylated CpG motifs in bacterial DNA or viruses (TLR9) [44]. Therefore, achieving a desired type of functional immune response, such as balanced Th1/Th2-mediated immunity, is of utmost importance. In this respect, we previously described a super-assembled and multimerized TLR7/8 agonist, R848, using a PCPP backbone (PCPP-R) and showed that it can dramatically boost the in vivo immune response to the HCV E2 protein [23]. In this study, we advanced this approach by synergizing the rationally designed BECC438 TLR4 agonist with PCPP-R to derive a novel binary adjuvant (PCPP-R+BECC438) and used the HCV sE2 antigen to compare immune responses against its individual components PCPP-R and BECC438 in BALB/c mice (Table 1). The mouse model has some inherent limitations, in particular, that it cannot be infected with HCV thereby precluding challenge studies. Moreover, mice lack orthologues of the human *IGHV1-69*-encoded bnAbs which are known to be important for protection against HCV infections. However, the mouse model is extensively used and well-suited to provide side-by-side comparisons between different groups in a given study prior to advancing to non-human primates, which are a limited and costly resource.

As shown in Figure 4a, immunization of mice with the binary adjuvant or the PCPP-R alone formulation produced an overall 10-fold higher anti-E2 IgG response compared to BECC438 alone or the no-adjuvant group. The magnitude of anti-E2 IgG titers remained similar between the PCPP-R and binary adjuvant groups (Figure 4a). Since PCPP-R is known to induce high IgG2a titers with HCV-sE2 [23], and IgG2a and IgG1 isotypes are commonly used as surrogate markers of Th1 and Th2 responses in mice, respectively, [45,46,47] we measured the impact of combining BECC438 with the PCPP-R adjuvant. We determined that the binary adjuvant produced a statistically significant 1.8-fold higher IgG2a/IgG1 ratio than PCPP-R alone, indicating that the binary adjuvant further improves the immune response to favor Th1 type immunity (Figure 4d). In HCV neutralization assays, all three adjuvants produced modest to high neutralization titers; however, individual mice in the binary adjuvant group showed higher geometrical mean titers than the other groups, such that the neutralization ID50 titer from the binary adjuvant group was 2–3 fold higher than BECC438 or PCPP-R group, respectively (Figure 5b), implying a potential synergy from combining PCPP-R+BECC438 as a single binary adjuvant.

## 5. Conclusions

Overall, our results provide a simple method that utilizes a PCPP scaffold to present multiple TLR agonists—a Lipid A-TLR4 agonist (BECC438) and small molecule-resiquimod—TLR7/8 as a single-stable adjuvant formulation. Such a combination adjuvant should help vaccine antigens with inherently poor immunogenicity such as HCV-envelope antigen. The aqueous formulation should ameliorate the injection site reactions associated with use of lipid-containing adjuvants (MF59, AS03) in vaccines [48]. Additionally, unlike lipid/oil emulsion adjuvants that suffer from phase separation from antigens upon longer-term storage [49,50], PCPP remains amenable to lyophilization and, upon resuspension can potentially maintain the stable complex with antigen and TLR-agonists while maintaining the desired particle size. Therefore, our approach should help hold the antigen and TLR agonist(s) in close proximity for optimal immune stimulation and facilitate stable long-term storage of the prepared vaccine formulation.

## Figures and Tables

**Figure 1 vaccines-13-01012-f001:**
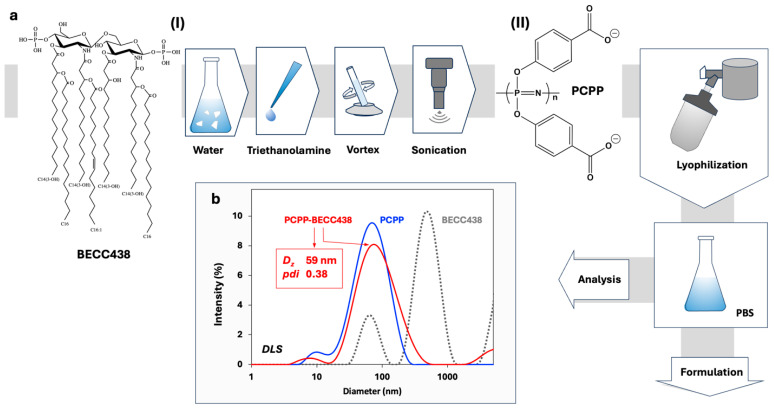
Preparation and characterization of PCPP-BECC438 formulations. (**a**) Flowchart of a multi-step preparation of aggregate-free PCPP-BECC438 formulations and (**b**) DLS profiles of BECC438, PCPP, and their formulation obtained using the process shown in panel a (0.25 mg/mL BECC438, 0.5 mg/mL PCPP, PBS, pH 7.4, z-average hydrodynamic diameter (*D_z_*) and polydispersity index (*pdi*) of lyophilized and redissolved PCPP-BECC438 formulations are shown; please see text for the description of steps (I) and (II)).

**Figure 2 vaccines-13-01012-f002:**
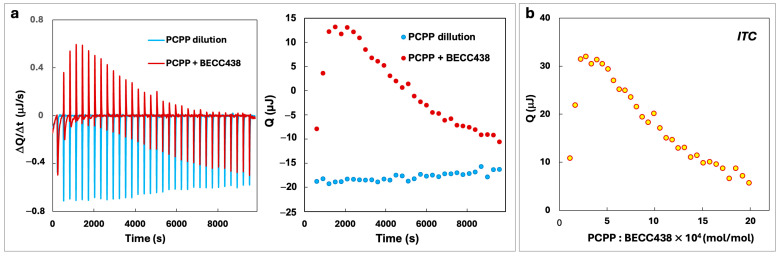
(**a**) ITC data of PCPP interaction with BECC438 and the corresponding heats of dilution of PCPP (50 mM phosphate buffer, pH = 7.4, 25 °C) and (**b**) binding isotherm corrected for the heat of dilution.

**Figure 3 vaccines-13-01012-f003:**
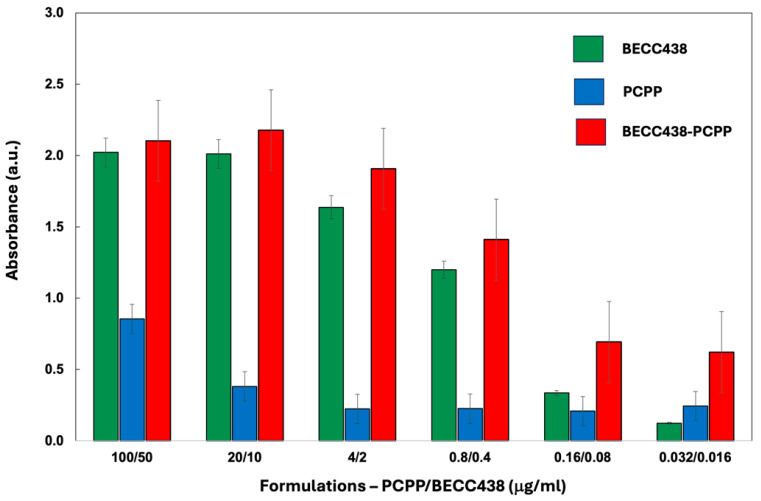
Immunoactivation of RAW BLUE cells by the PCPP-BECC438, BECC438, and PCPP formulations as measured by the spectrophotometric monitoring of SEAP substrate hydrolysis (all formulations contain HCV E2 antigen (100 μg), the absorbance measured at 405 nm, *n* = 3, error bars—standard error).

**Figure 4 vaccines-13-01012-f004:**
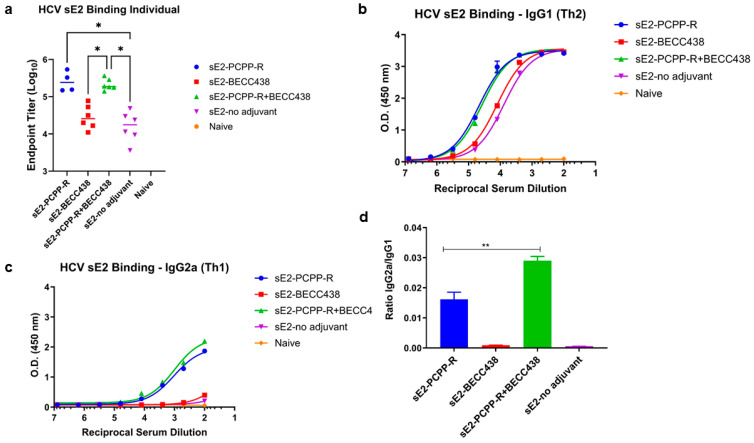
Assessment of immunogenicity induced by PPZ-R and BECC adjuvanted formulations by an ELISA in mice at day 56 after immunization. (**a**) Total IgG titers, (**b**) total IgG1 titers, (**c**) total IgG2 titers, and (**d**) the ratio of IgG2a to IgG1 for each immunogen. Individual mouse sera were analyzed. Endpoint titers were calculated by curve fitting in GraphPad software (version 10) with endpoint optical density defined as four times the highest absorbance value of preimmune sera. The experiments were performed in duplicate. Error bars represent the standard deviation of the data points. The statistical significance of differences in endpoint titers was evaluated using Kruskal–Wallis analysis of variance with Dunn’s multiple comparison test. Statistical significance of differences in IgG2a/IgG1 ratios was evaluated using a one-way analysis of variance (ANOVA) test. (* *p* < 0.05, ** *p* < 0.01).

**Figure 5 vaccines-13-01012-f005:**
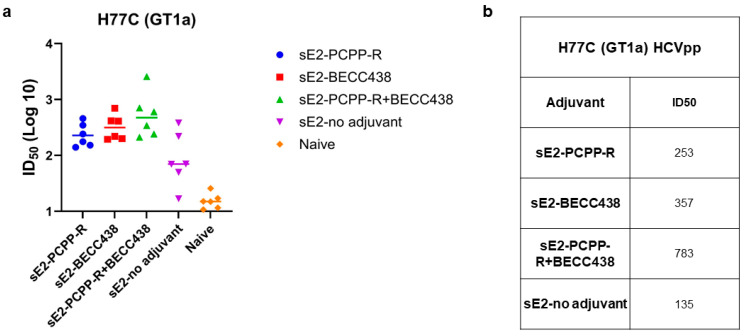
Potency of neutralization of individual mouse sera against H77C (GT1a) isolates, comparing PCPP-R and BECC438 formulations. (**a**) Individual sera were assessed for neutralization activities at day 56 and day 0. Serum dilutions were performed as three-fold dilutions starting at 1:150 for HCVpp neutralization, and each experiment was performed in duplicate. (**b**) Inhibition dilution values correspond to 50 percent neutralization (ID50) values for each group. Statistical analyses using Kruskal–Wallis analysis of variance with Dunn’s multiple comparison test did not detect significant differences between the groups.

**Table 1 vaccines-13-01012-t001:** The design of in vivo studies.

Group	Label	sE2 FL (µg)	PCPP-R (µg)	BECC438 (µg)
1	sE2-PCPP-R	50	50	0
2	sE2-BECC438	50	0	25
3	sE2-PCPP-R+BECC438	50	50	25
4	sE2-no adjuvant	50	0	0
5	Naive	0	0	0

## Data Availability

The original contributions presented in this study are included in the article. Further inquiries can be directed to the corresponding authors.

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
