# Peer review of "Polyphosphazene-Mediated Assembly of TLR4 and TLR7/8 Agonists Enables a Potent Nano-Adjuvant Delivery System for Hepatitis C Virus Vaccine Antigens"

_vaccines, 2025, doi:10.3390/vaccines13101012_

Round 1

Reviewer 1 Report

Comments and Suggestions for Authors

In this manuscript by A.K. Andrianov and colleagues, the authors used a combination of a lipid A Toll-like receptor 4 (TLR4) agonist BECC438 and TLR7/8 agonist resiquimod (R848) on a polyphosphazene macromolecule (PCPP) to create a nanoscale immunoadjuvant-vaccine (PCPP-R+BECC438) for the delivery of the hepatitis C virus (HCV) E2 glycoprotein. This formulation was used to assess immune response in vivo and compared to the single adjuvants.

Besides considering this work for publication, I would recommend the following revisions:

  • The authors assessed the antibody response against the H77 HCV E2 and HCVpp only. It would be interesting to explore the breadth of recognition and functional activity beyond this strain and genotype 1a.
  • In addition to HCVpp, the authors should perform and confirm the neutralization against the HCV cell culture system.
  • No details about the HCV E2 glycoprotein construct is provided. Details should be added and specify the strain and genotype in the material and methods section and not mentioning it just in the following sections.
  • The authors did not evaluate the T cell response, which plays a crucial role in conferring protection and clearance upon HCV infection.
  • How does the use of these adjuvants experimentally compare with the use of other commonly approved or under development adjuvants (MF59, AS01, AS03, and CpG 1018)

Author Response

In this manuscript by A.K. Andrianov and colleagues, the authors used a combination of a lipid A Toll-like receptor 4 (TLR4) agonist BECC438 and TLR7/8 agonist resiquimod (R848) on a polyphosphazene macromolecule (PCPP) to create a nanoscale immunoadjuvant-vaccine (PCPP-R+BECC438) for the delivery of the hepatitis C virus (HCV) E2 glycoprotein. This formulation was used to assess immune response in vivo and compared to the single adjuvants.

Besides considering this work for publication, I would recommend the following revisions:

  1. The authors assessed the antibody response against the H77 HCV E2 and HCVpp only. It would be interesting to explore the breadth of recognition and functional activity beyond this strain and genotype 1a.

Response: We tested the ability of the mouse sera to neutralize heterologous HCVpp, but the neutralization was very weak for all groups and therefore we didn’t observe any differences between the groups. This study was focused more on a proof-of-principle demonstration that R848, BECC438, and the antigen (in this case HCV E2) could all be co-formulated and elicit a strong homologous immune response in mice. It should be noted that we have a study currently underway that explores the issue of cross-neutralization in more depth and which will be the subject of a future publication.

  1. In addition to HCVpp, the authors should perform and confirm the neutralization against the HCV cell culture system.

Response: Studies have shown a positive correlation between neutralization data obtained using HCVpp and HCVcc [PMID: 27667373, 26699643], which would indicate that HCVcc neutralizations should yield the same result. We are in the process of looking for a collaborator with the relevant capabilities. However, the outcome of such studies, which will likely to take months to be completed, will not alter the main conclusion of the manuscript, which is largely focused on the nano-formulation approach and in vivo results.

  1. No details about the HCV E2 glycoprotein construct is provided. Details should be added and specify the strain and genotype in the material and methods section and not mentioning it just in the following sections.

Response: We have added a brief description of the construct as requested to the Materials and Methods section (lines 132-135).

  1. The authors did not evaluate the T cell response, which plays a crucial role in conferring protection and clearance upon HCV infection.

Response: Although T-cell responses were not evaluated for formulations of the present Study, the isotype-specific analysis was conducted with IgG2a (a surrogate marker for the strength of the cellular immune response) assessed. Further studies are planned, in which T-cell responses will be evaluated for different variations of the nanoscale formulation to establish a contribution or TLR4 and TLR7/8 agonists. It can be also noted that T-cell response to BECC438 adjuvanted vaccines are well-established (please see (a) 10.1016/j.vaccine.2024.126577; (b) 10.3389/fimmu.2024.1372349; (c)1101/2022.12.07.519460). This adjuvant potency is also reviewed briefly in Reference 9 of the manuscript.

  1. How does the use of these adjuvants experimentally compare with the use of other commonly approved or under development adjuvants (MF59, AS01, AS03, and CpG 1018)

Response: In contrast with most adjuvants above (MF59, AS01, AS03), combination adjuvants of the present paper present a water-soluble nanoscale range system, which is easy to lyophilize and formulate. Comparison of newly developed adjuvants with commercially produced adjuvants listed above is generally challenging due to limited availability of the latter and restrictions on publications imposed by the manufacturer. Some comparative studies, however, were performed and the results for polyphosphazene adjuvants are incorporated in this manuscript by Reference [11]. As such, a comparison of PCPP with another water-soluble CpG showed a superior performance of PCPP in a number of studies (Reference 11). Studies on the comparison of BECC438 with AS01 are under way and will be reported timely.

Reviewer 2 Report

Comments and Suggestions for Authors

The authors address an important topic about Polyphosphazene-Mediated Assembly of TLR4 and TLR7/8 Agonists Enables a Potent Nano-Adjuvant Delivery System for Hepatitis C Virus Vaccine Antigens. Despite this some points should be addressed before publishing.

Introduction should be contains more information about TLR4  as PAMP member and thier role in immune system. Moreover , add examples of TLR4 agonists. Please add information about animals model used for study HCV.

Methods, please add details about characterization of prepared  nanoscale vaccines delivery systems. Moreover, add the rationale of using mice for in vivo study.

Results, each figure legend should be contains information about statistical analysis data expression SEM or SD, sample number, p value and significant limit.

Discussion please support your work with similar finding and add the contradictory if present. Please, discussion in detail, how the components of the selected system were assembled in nanoformulation and thier similarities with HCV? Please give details about their interaction with imune cells.Moreover, discuss the limitations of mice model in HCV. However there are genetic differences compared with humans. Moreover, discuss the study limitations in general.

Please, check the manuscript for very long sentences or paragraphs without references citation.

Please check the references list for 2025 citation dated.

Comments on the Quality of English Language

Please, check the manuscript for minor grammar edition and syntax.

Author Response

  1. The authors address an important topic about Polyphosphazene-Mediated Assembly of TLR4 and TLR7/8 Agonists Enables a Potent Nano-Adjuvant Delivery System for Hepatitis C Virus Vaccine Antigens. Despite this some points should be addressed before publishing. Introduction should be contains more information about TLR4  as PAMP member and thier role in immune system. Moreover , add examples of TLR4 agonists. Please add information about animals model used for study HCV.

Response: This information was added to the first paragraph of the Introduction.

  1. Methods, please add details about characterization of prepared nanoscale vaccines delivery systems. Moreover, add the rationale of using mice for in vivo.

Response: Details on characterization of nanoscale vaccine formulations are now added to section 2.5. We have also added language regarding the use and limitations of the mouse model in the discussion section.

  1. Results, each figure legend should be contains information about statistical analysis data expression SEM or SD, sample number, p value and significant limit.

Response: We have added the requested information to the figure legends.

  1. Discussion please support your work with similar finding and add the contradictory if present. Please, discussion in detail, how the components of the selected system were assembled in nanoformulation and thier similarities with HCV? Please give details about their interaction with imune cells. Moreover, discuss the limitations of mice model in HCV. However there are genetic differences compared with humans. Moreover, discuss the study limitations in general.

Response: Details on nanoformulation, interactions with immune cells and similarity of synthetic nano-assemblies with viruses were added to the end of the first paragraph of the Discussion section. We have also added language regarding the limitations of the mouse model in the discussion section.

  1. Please, check the manuscript for very long sentences or paragraphs without references citation.

Response: We reviewed the manuscript and rephrased multiple sentences.

  1. Please check the references list for 2025 citation dated.

Response: Reference 17 was checked and information was found to be correct.

Reviewer 3 Report

Comments and Suggestions for Authors
  • For Figure 3, can you comment on the immunoactivation of RAW Blue cells that were incubated with HCV E2 antigen without PCPP or BECC438? I'm curious if the antigen itself can activate the cells due to something like endotoxin contaminants. Also a bit unclear how much antigen is incubated with the cells.
  • In vitro stimulation of macrophages or dendritic cells and quantifying cytokine production would be interesting to see between PCPP, PCPP-R, BECC438, resiquimod, PCPP+BECC438, and PCPP-R+BECC438. Could elucidate any synergistic effects. This would may also support the IgG2a production differences between groups in the in vivo study.

Author Response

For Figure 3, can you comment on the immunoactivation of RAW Blue cells that were incubated with HCV E2 antigen without PCPP or BECC438? I'm curious if the antigen itself can activate the cells due to something like endotoxin contaminants. Also a bit unclear how much antigen is incubated with the cells.

Response: We were unable to observe any activation without adjuvants. Please note that even the response from antigen-PCPP formulation was also minimal (Figure 3). For clarity, the amount of antigen was specified in the legend to Figure 3 (100 mg).

In vitro stimulation of macrophages or dendritic cells and quantifying cytokine production would be interesting to see between PCPP, PCPP-R, BECC438, resiquimod, PCPP+BECC438, and PCPP-R+BECC438. Could elucidate any synergistic effects. This would may also support the IgG2a production differences between groups in the in vivo study.

Response: Thank you for this valuable advice. We certainly plan such experiments with dendritic cells and are currently looking for a reliable source for them. This will be the subject of our future publications. As for PCPP activation of dendritic cells, please see our previous publication: C.D. Palmer, J. Ninković, Z.M. Prokopowicz, C.J. Mancuso, A. Marin, A.K. Andrianov, D.J. Dowling and O. Levy, The effect of stable macromolecular complexes of ionic polyphosphazene on HIV Gag antigen and on activation of human dendritic cells and presentation to T-cells. Biomaterials, 2014, 35 (31), pp. 8876-86. DOI: 10.1016/j.biomaterials.2014.06.043. PMID: 25023392.

Round 2

Reviewer 3 Report

Comments and Suggestions for Authors

N/A

Author Response

Thank you for the valuable advice.